# The Discovery of Indole-2-carboxylic Acid Derivatives as Novel HIV-1 Integrase Strand Transfer Inhibitors

**DOI:** 10.3390/molecules28248020

**Published:** 2023-12-08

**Authors:** Yu-Chan Wang, Wen-Li Zhang, Rong-Hong Zhang, Chun-Hua Liu, Yong-Long Zhao, Guo-Yi Yan, Shang-Gao Liao, Yong-Jun Li, Meng Zhou

**Affiliations:** 1State Key Laboratory of Functions and Applications of Medicinal Plants, Engineering Research Center for the Development and Application of Ethnic Medicine and TCM (Ministry of Education), Guizhou Medical University, Guiyang 550004, China; wyc1026508@163.com (Y.-C.W.); 18083242928@163.com (W.-L.Z.); 15186951394@163.com (R.-H.Z.); liyongjun026@126.com (Y.-J.L.); 2Center for Tissue Engineering and Stem Cell Research, Key Laboratory of Regenerative Medicine of Guizhou Province, School of Basic Medical Sciences, Guizhou Medical University, Guiyang 550004, China; 3School of Pharmacy, Guizhou Medical University, Guian New District, Guiyang 550025, China; zhaoyl05@126.com (Y.-L.Z.); lshangg@163.com (S.-G.L.); 4School of Pharmacy, Xinxiang University, Xinxiang 453000, China; ygyxyz307@hotmail.com

**Keywords:** HIV-1 integrase, design and synthesis, indole-2-carboxylic acid, virtual screening, structural optimization

## Abstract

As an important antiviral target, HIV-1 integrase plays a key role in the viral life cycle, and five integrase strand transfer inhibitors (INSTIs) have been approved for the treatment of HIV-1 infections so far. However, similar to other clinically used antiviral drugs, resistance-causing mutations have appeared, which have impaired the efficacy of INSTIs. In the current study, to identify novel integrase inhibitors, a set of molecular docking-based virtual screenings were performed, and indole-2-carboxylic acid was developed as a potent INSTI scaffold. Indole-2-carboxylic acid derivative **3** was proved to effectively inhibit the strand transfer of HIV-1 integrase, and binding conformation analysis showed that the indole core and C2 carboxyl group obviously chelated the two Mg^2+^ ions within the active site of integrase. Further structural optimizations on compound **3** provided the derivative **20a**, which markedly increased the integrase inhibitory effect, with an IC_50_ value of 0.13 μM. Binding mode analysis revealed that the introduction of a long branch on C3 of the indole core improved the interaction with the hydrophobic cavity near the active site of integrase, indicating that indole-2-carboxylic acid is a promising scaffold for the development of integrase inhibitors.

## 1. Introduction

The high morbidity and mortality associated with acquired immune deficiency syndrome (AIDS), which is caused by human immunodeficiency virus (HIV-1), make this pandemic a major global public health problem [1,2,3]. HIV-1 integrase plays a critical role in viral replication, inserting the double-stranded viral DNA, generated from reverse transcription, into the host genome [4,5], thus leading to the establishment of proviral latency [6]. As an important protein in viral infection, integrase is a particularly attractive antiviral target because there is no host cellular counterpart, and specific integrase inhibitors would not interfere with cellular functions [7]. In addition, since integrases use the same active site (DDE motif) for both the 3′-processing and DNA strand transfer steps, inhibitors targeting integrase benefit from a potentially high genetic barrier to resistance selection [8]. Therefore, extensive efforts have been made to identify effective integrase inhibitors with diverse chemical features [9,10,11,12].

A diketo acid moiety was believed to be essential for the development of integrase inhibitors [13,14,15], and, so far, five integrase antagonists have been approved for clinical use: raltegravir (RAL, Figure 1) [16], elvitegravir (ELV) [17], dolutegravir (DTG) [18], bictegravir (BIC) [19] and cabotegravir (CAB) [20]. All these integrase strand transfer inhibitors (INSTIs) effectively inhibit the integration of DNA catalyzed by integrase [21,22]. However, drug resistance mutations inevitably emerged during the antiviral therapy and impaired the susceptibility of the virus to the first-generation inhibitors (RAL and ELV) [19,23,24]. The use of the three-ring system in second-generation inhibitors (DTG, BIC and CAB) enhanced the potency against some but not all RAL/EVG resistant HIV variants [25,26]. Therefore, the development of new integrase inhibitors with novel structural scaffolds is still an important research topic.

Since the high-resolution cryo-electron microscopy (cryo-EM) structures of HIV intasome were reported, the mode of action of advanced INSTIs has been well analyzed, including a chelating core with two Mg^2+^ of integrase and a π-stacking interaction with 3′ terminal adenosine of processed vDNA (dC20) [27,28]. It should be noted that a hydrophobic cavity near the β4-α2 connector was formed by the sidechain of Tyr143 branch and the backbone of Asn117, which is close to the active site of integrase [10,27]. Although the three-ring system associated with the second-generation inhibitors DTG and BIC extended to the location, substantial free space was still observed in this cavity. Thus, improving the interaction with the hydrophobic site might increase the antiviral effects of INSTIs and delay the emergence of drug-resistant mutations.

Herein, a structure-based virtual screening was carried out to find novel chemical scaffolds chelating two Mg^2+^ cofactors within the active site of integrase (Figure 1). Halogenated phenyls were then added to the chemical structure to form a π-π stacking interaction with dC20, as in previous research [29]. Most importantly, to fill the space in the hydrophobic cavity, a bulky hydrophobic pharmacophore was then introduced to the core of the optimized structure through a linker. Sequentially, structural optimizations were performed to complete SAR study. Through these efforts, potent and effective INSTIs were developed, and the binding mode of the hydrophobic site was also analyzed.

## 2. Results and Discussion

### 2.1. Molecular Docking-Based Virtual Screening

As a rapid and flexible docking module, LibDock was developed as a site feature algorithm for docking small molecules into an active receptor site [30,31], which has been proved to be an effective tool for the discovery of potential chemical entities [32,33,34]. In order to develop potent HIV-1 integrase inhibitors, LibDock-based virtual screening was conducted to find potent INSTIs in this study. The Chemdiv and Specs databases, which include 1.75 million compounds, were selected to be docked into the crystal structure of HIV-1 integrase (PDB ID: 6PUY), and the location of the original substrate (4d) was designated as the docking site [28]. After being sorted by LibDock score in Discovery Studio 3.1, the 175 compounds with highest docking scores were retained. Furthermore, to eliminate molecules with poor drug-like properties, all the selected compounds were then filtered using Lipinski’s rules [35], providing a total of 168 compounds (listed in Appendix A).

Subsequently, the Autodock Vina program was used to investigate the binding energy of the above 168 compounds with HIV-1 integrase, and the 25 compounds with the lowest binding energy (Appendix A) were selected. Chelating with the metal ions of integrase is critical for INSTIs [27,36], and compounds with obvious metal chelating properties and novel structural scaffolds were retained. Therefore, a total of 10 compounds (**1**–**10**, Figure 2) were finally left, and the binding modes of these complexes are depicted in Appendix A. The binding results revealed that all these compounds effectively bound with the active site of HIV-1 integrase, with binding energy lower than −12.9 kcal/mol (Table 1). Meanwhile, for most of the compounds (**1**–**5** and **10**), the energy needed for the most frequent conformation also manifested in the lowest energy, indicating the reliability of molecular docking results. Specifically, obvious metal-chelating interactions were observed for all 10 compounds, and π-π stacking interactions were formed with dC20 by eight of them (except **3** and **4**, Appendix A). Thus, these compounds were further evaluated for their inhibitory effect against HIV-1 integrase.

### 2.2. Evaluation of Integrase Strand Transfer Inhibitory Effect

DNA strand transfer is essential for viral replication, and all the clinically used HIV integrase antagonists are INSTIs [37]. Thus, the integrase strand transfer-inhibitory activity was tested for the above 10 compounds using an HIV-1 integrase assay kit, and **RAL** was used as the positive control [38,39] (Table 1). The results showed that all the molecules clearly inhibited the strand transfer of HIV-1 integrase with IC_50_s from 12.41–47.44 μM, suggesting the effective hit rates of the former virtual screening. Among them, compounds **3** and **4** manifested the highest integrase inhibition activity, with IC_50_s of 12.41 and 18.52 μM, respectively. Interestingly, all the clinically used INSTIs effectively interacted with dC20 through a π-π stacking effect; however, this interaction was not observed for compounds **3** and **4**, demonstrating the potential for further optimizations. Through analyzing the binding mode of the two compounds, it was found that a bis-bidentate chelation of the Mg^2+^ ions was established for the chromeno core and 2-isoxazole of **4**, and a π-π stacking interaction with dA21 was also found (Figure 3A). For compound **3**, a chelating triad motif was found between the indole-2-carboxylic acid scaffold and the two metals within the active site, and a hydrophobic interaction was observed between the indole core and dA21 (Figure 3B). Moreover, the C3 carboxyl group extended to the hydrophobic cavity near the β4-α2 connector, indicating that a structural modification in this position might improve the antiviral activity. Considering the pronounced antiviral activity and potential of the core structure, compound **3** was selected for further optimizations. 

### 2.3. Optimization Strategies

Through comparing the binding conformation of compound **3** and the second-generation HIV-1 integrase inhibitors [27,40], we speculated that a π-stacking interaction with viral DNA would be formed through the addition of a halogenated phenyl to C6 of the indole core of compound **3** (Figure 4). Meanwhile, we speculated that modifications of the C2 carboxyl group will also be carried out to improve the chelation with metal ions of the intasome core. Most importantly, to enhance the antiviral effect of INSTIs and delay the emergence of drug-resistant mutations, interactions with the hydrophobic cavity near the integrase active site will then be investigated [10,27]. Thus, the length and size of the C3 substituent will be modified to improve the interaction with the backbone residues that form the hydrophobic pocket, such as Tyr143 and Asn117.

### 2.4. Synthesis route of Compound **3** Derivatives

To facilitate the introduction of halogenated phenyl groups at the C6 position of the indole core, 6-bromoindole-2-carboxylic acid (**11**) was used as the initial material, and the synthetic route is described in Figure 1. The carboxyl group of **11** was esterificated to produce compound **12** in the presence of concentrated sulfuric acid. Due to the electron absorption effect of the C2 carbonyl group, a formyl group was introduced at the C3 position through the Vilsmeier–Haack reaction to provide a high yield of compound **13** (87%), which was then selectively reduced to hydroxymethyl (**14**) by using isopropanolic aluminium in the Meerwein–Ponndorf–Verley reaction. Interestingly, an ester-exchange product (**14**) was detected under this condition. Sequentially, compound **14** was condensed with *p*-(trifluoromethyl)benzyl alcohol and *o*-fluorobenzyl alcohol to provide compounds **15** and **18** under alkaline conditions, respectively. For compound **15**, 3-fluoro-4-methoxyphenyl and 2,4-difluorophenyl were introduced at C6 via the Buchwald–Hartwig reaction catalyzed by palladium acetate to offer compounds **16a** and **16b**, which were then hydrolyzed to produce **17a** and **17b**, respectively. Similarly, halogenated phenyls were added to the C6 of compound **18** to provide **19a** and **19b** as well, which were further hydrolyzed to provide **20a** and **20b** in the presence of alkaline conditions, respectively.

### 2.5. Evaluation of Biological Activity

The inhibitory effect of compound **3** derivatives against integrase strand transfer was then evaluated. As shown in Table 2, all the synthesized compounds exhibited better integrase inhibitory activities than the parent compound, with IC_50_s from 0.13 to 6.85 μM. Encouragingly, the introduction of long-chain *p*-trifluorophenyl (**15**) or *o*-fluorophenyl (**18**) at the C3 position of the indole core indeed significantly improved the activity by 5.3-fold and 6.5-fold, respectively. Meanwhile, the addition of halogenated anilines at the C6 position (**16b**, **19a** and **19b**) markedly improved the integrase inhibition effect, with IC_50_s of 1.05–1.70 μM, probably due to the formation of π-π stacking with viral DNA. Moreover, further hydrolysis of the carboxylate of **16a**, **16b**, **19a** and **19b** resulted in clear enhancements in activity. This was especially clear for compounds **17b**, **20a** and **20b**, as the IC_50_ values were 0.39, 0.13 and 0.64 μM, respectively. The remarkable improvement in inhibition was probably due to the increase in the bis-bidentate chelation between the Mg^2+^ ions and the free carboxyl group. Among them, the inhibitory effect of **20a** was close to that of the positive control RAL (0.06 μM). 

On the other hand, the toxicity of the synthesized compounds to the MT-4 human T-cell line was also assessed (Table 2). The results revealed that no toxicity was observed for most of the compounds at 80 μM except for **18**, **19a** and **19b**, and the CC_50_ values of these three compounds were all higher than 29 μM, indicating lower toxicities.

### 2.6. Binding Mode Analysis

To better understand the structure–activity relationship, the three compounds with best integrase inhibitory effect (**17b**, **20a** and **20b**) were selected and further studied for their binding modes with HIV-1 integrase using molecular docking (Figure 5). The results revealed that compounds **17b**, **20a** and **20b** bound to the integrase in a similar binding conformation. Specifically, three central electronegative heteroatoms were observed to chelate with two Mg^2+^ cofactors within the active site, indicating a strong metal-chelating cluster between the C2 carboxyl indole with the integrase. Furthermore, in agreement with the previous design, π-π stacking interactions were also found in the introduced C6 halogenated benzene with dC20, which increased the binding force with the integrase.

The reason for the difference in activity of compounds **17b**, **20a** and **20b** was then investigated, and it was decided that this was probably because of the long branches at the C3 position of the indole core that extended to the hydrophobic cavity formed by the sidechain of Tyr143 branch and the backbone of Asn117 [10,27]. Compared with compound **3**, two hydrogen bonds were observed between the trifluoromethyl group of **17b** and Asn117 (Figure 5A), which might contribute to the significant enhancement in the integrase-inhibitory effect. For compound **20a**, a similar H-bond was also found (Figure 5B). Moreover, a critical π-π stacking interaction was formed between the benzene of the C3 branch and Tyr143, resulting in an improvement in activity. In contrast, although the above π-π stacking interaction was partially retained in compound **20b**, the H-bond with Asn117 disappeared because of a 180° flip in the benzene of the C3 branch, leading to the decline in activity (Figure 5C).

## 3. Conclusions

In summary, to identify new HIV-1 integrase inhibitors, a set of molecular docking-based virtual screenings were carried out, and 10 molecules were chosen for biological activity study, which obviously inhibited the strand transfer of integrase. Among these compounds, a chelating core with two Mg^2+^ of integrase was observed in the indole core and the C2 carboxyl group of compound **3**. Through optimizations at C2, C3 and C6 of the indole core of compound **3**, a series of indole-2-carboxylic acid derivatives were designed and synthesized, and the biological results showed that all synthesized compounds markedly improved the inhibitory effect against integrase. Furthermore, the structure–activity relationship revealed that introducing C6-halogenated benzene and the C3 long branch to the indole core indeed increased the inhibitory activity against HIV-1 integrase. Through the above structural modifications, compound **20a** was finally developed, which significantly inhibited the strand transfer of integrase with IC_50_ of 0.13 μM. Binding mode analysis suggested that the C3 long branch of **20a** extended to the hydrophobic cavity near the integrase active and interacted with Tyr143 and Asn117. The above results demonstrated the potential of indole-2-carboxylic acid derivative **20a** as a novel HIV-1 INSTI. 

## 4. Experimental Section

### 4.1. General Methods and Materials

All analytical grade reagents and chemicals were purchased from commercial suppliers. Column chromatography was performed on silica gel (Qingdao, 300–400 mesh) using the indicated eluents. ^1^H NMR and ^13^C NMR spectra were recorded on a JEOL spectrometer (400 MHz) using CDCl_3_, CD_3_OD or DMSO-*d*_6_ as solvents with the internal standard of tetramethylsilane. HRMS was recorded on a Thermo Scientific Q Exactive Plus Orbitrap LC-MS/MS. All prepared compounds were purified to >96% purity as determined by HPLC (Dionex Ultimate 3000, Thermo Fisher Scientific, Waltham, MA, USA) analysis using the following methods. Purity analysis of final compounds was performed through a SuperLu C_18_ (particle size = 5 μm, pore size = 4.6 nm, dimensions = 250 mm) column. The injection volume was 10 μL, the mobile phase consisted of methyl alcohol and water (80:20) with a flow rate of 1.0 mL/min and each analysis lasted for 20 min. The detection wavelength was 280 nm. The retention time of each compound (R_T,HPLC_) was displayed in the analytical data of the respective compounds. All melting points were obtained with a WRS-2 microcomputer melting point apparatus.

### 4.2. LibDock Based Virtual Screening

LibDock based virtual screening was used to identify potential HIV-1 integrase inhibitors. The X-ray crystal structure of HIV-1 integrase (PDB ID: 6PUY [28]) was downloaded from the RCSB Protein Data Bank (www.rcsb.org, accessed on 16 December 2022). Then, 1.75 million compounds from the Chemdiv and Specs databases were docked into the crystal structure of integrase using the LibDock module in the Discovery Studio software (3.1); the location of the ligand (4d) was selected as the binding site, 50 conformations were generated for each molecule and the other parameters were set to default. The 175 compounds (1/10,000) with the highest LibDock scores were retained, which were then filtered using the ‘Lipinski’s rule of five’ [41] in Discovery Studio.

### 4.3. Molecular Docking

Molecular docking of 168 compounds with drug-like property was conducted using Autodock Vina to predict the binding affinity as well as binding modes of the compounds with integrase. The location of the ligand (4d) was used as the binding site with a central coordinate in the X (141.136), Y (159.768), and Z directions (179.84). All the rotatable bonds in the ligands were allowed to rotate freely, and 100 conformations were generated for each molecule. After docking, the best conformation of each ligand was selected and the corresponding score of the ligand was ranked to find the compounds with the highest score. The top 25 compounds with the highest scores were selected for PAINS remover and manual analysis.

### 4.4. PAINS Remover

False-positive problems might occur during the virtual screening, and PAINS (Pan Assay Interference Compounds) is a compound that interferes with most tests. To avoid the possible false-positive problems in subsequent biological tests, PAINS remover (https://www.cbligand.org/PAINS/, accessed on 2 February 2023) was used to remove the possible PAINS molecule [42].

### 4.5. General Procedure for the Synthesis of Compounds **12**–**21**

#### 4.5.1. Synthesis of Ethyl 6-bromo-1*H*-indole-2-carboxylate (**12**)

6-bromo-1*H*-indole-2-carboxylic acid (100 mg, 0.42 mmol) was dissolved in alcohol (10 mL), and concentrated sulfuric acid (20 mg, 0.21 mmol) was added dropwise to the solution. The mixture was stirred at 80 °C for 2 h (monitored by TLC). The reaction was quenched with a saturated aqueous solution of sodium bicarbonate and extracted with ethyl acetate (30 mL × 3). The combined organic phase was dried over anhydrous sodium sulfate before vacuum suction filtration. The crude product was chromatographed on silica gel (1:5 *v*/*v* ethyl acetate/petroleum ether) to afford a white solid (91 mg, 82%). ^1^H NMR (400 MHz, CDCl_3_), *δ* 9.19 (br, 1H), 7.60 (s, 1H), 7.54 (d, *J* = 8.4 Hz, 1H), 7.24 (s, 1H), 7.19 (s, 1H), 4.43 (q, *J* = 7.2 Hz, 2H), 1.43 (t, *J* = 6.8 Hz, 3H). ^13^C NMR (100 MHz, CDCl_3_) *δ* 161.81, 137.38, 128.07, 126.27, 124.37, 123.81, 119.07, 114.77, 108.62, 61.29, 14.38. ESI-HRMS (*m*/*z*): calcd C_11_H_10_BrNO_2_ for [M − H]^−^ 265.9817, 267.9796; found 265.9822, 267.9798. R_T,HPLC_ = 12.51 min.

#### 4.5.2. Synthesis of Ethyl 6-bromo-3-formyl-1*H*-indole-2-carboxylate (**13**)

Phosphorus oxychloride (572 mg, 3.73 mmol) was added dropwise to a solution of compound **12** (100 mg, 0.37 mmol) in DMF (10 mL). The mixture was stirred at room temperature for 2 h and sequentially heated under reflux for 2 h (monitored by TLC). The solution was cooled to room temperature, brought to pH8 by adding anhydrous sodium carbonate, extracted with ethyl acetate (30 mL × 3) and concentrated under reduced pressure to afford the crude product. The residue was applied to silica gel column chromatography (1:4 *v*/*v* ethyl acetate/petroleum ether) to provide a white solid (96 mg, 87%). ^1^H NMR (400 MHz, DMSO-*d_6_*) *δ* 12.82 (s, 1H), 10.58 (s, 1H), 8.17 (d, *J* = 8.4 Hz, 1H), 7.72 (s, 1H), 7.43 (d, *J* = 8.4 Hz, 1H), 4.47 (q, *J* = 7.2 Hz, 2H), 1.41 (t, *J* = 7.2 Hz, 3H). ^13^C NMR (100 MHz, DMSO-*d_6_*) *δ* 187.92, 160.44, 137.09, 133.86, 127.02, 124.69, 124.23, 119.11, 118.87, 116.15, 62.53, 14.58. ESI-HRMS (*m*/*z*): calcd C_12_H_10_BrNO_3_ for [M − H]^−^ 293.9766, 295.9746; found 293.9822, 295.9795. R_T,HPLC_ = 8.37 min.

#### 4.5.3. Synthesis of Isopropyl 6-bromo-3-(hydroxymethyl)-1*H*-indole-2-carboxylate (**14**)

The synthesized intermediate **13** (100 mg, 0.34 mmol) and aluminum isopropoxide (207 mg, 1.01 mmol) were dissolved in isopropanol (15 mL) and stirred at 60 °C for 5 h (monitored by TLC). After removal of isopropanol using a rotary evaporator, the reaction mixture was applied to silica gel column chromatography (1:5 *v*/*v* ethyl acetate/petroleum ether) to provide a white solid (82 mg, 85%). ^1^H NMR (400 MHz, DMSO-*d_6_*) *δ* 11.59 (s, 1H), 7.83 (d, *J* = 8.8 Hz, 1H), 7.59 (s, 1H), 7.18 (d, *J* = 8.8 Hz, 1H), 5.21–5.14 (m, 1H), 4.97 (s, 2H), 4.86 (d, *J* = 5.6 Hz, 1H), 1.36 (d, *J* = 6.4 Hz, 6H). ^13^C NMR (100 MHz, DMSO-*d_6_*) *δ* 187.92, 160.44, 137.09, 133.86, 127.02, 124.69, 124.23, 119.11, 118.87, 116.15, 62.53, 14.58. ESI-HRMS (*m*/*z*): calcd C_13_H_14_BrNO_3_ for [M − H]^−^ 310.0079, 312.0059; found 310.0086, 312.0061. R_T,HPLC_ = 12.93 min.

#### 4.5.4. Synthesis of Isopropyl 6-bromo-3-(((4-(trifluoromethyl)benzyl)oxy)methyl)-1*H*-indole-2-carboxylate (**15**)

Compound **14** (100 mg, 0.32 mmol), 4-(trifluoromethyl)benzyl bromide (76 mg, 0.32 mmol) and potassium carbonate (133 mg, 0.96 mmol) were dissolved in DMF (10 mL). The solution was stirred at room temperature for 5 h (monitored by TLC). After the addition of 5 mL of water, the mixture was extracted with ethyl acetate (30 mL × 3), concentrated under reduced pressure to provide the crude product and applied to silica gel column chromatography (1:5 *v*/*v* ethyl acetate/petroleum ether) to afford liquid-oily products (133 mg, 93%). ^1^H NMR (400 MHz, DMSO-*d_6_*) *δ* 7.95 (d, *J* = 8.8 Hz, 1H), 7.89 (s, 1H), 7.67 (d, *J* = 7.6 Hz, 2H), 7.30 (d, *J* = 8.8 Hz, 1H), 7.14 (d, *J* = 7.6 Hz, 2H), 5.86 (s, 2H), 5.08 (t, *J* = 5.6 Hz, 2H), 5.00 (s, 2H), 1.20 (d, *J* = 6.0 Hz, 6H). ^13^C NMR (100 MHz, DMSO-*d_6_*) *δ* 161.36, 143.86, 139.30, 127.18, 125.93, 125.89, 125.66, 125.60, 125.40, 124.56, 124.06, 119.13, 114.00, 69.19, 55.71, 47.90, 21.92. ESI-HRMS (*m*/*z*): calcd C_21_H_19_BrF_3_NO_3_ for [M + Na]^+^ 492.0398, 494.0378; found 492.0389. 494.0364. The same procedure was also followed for the synthesis of **18**. 

#### 4.5.5. Synthesis of Isopropyl 6-((3-fluoro-4-methoxyphenyl)amino)-3-(((4-(trifluoromethyl)benzyl) oxy)methyl)-1*H*-indole-2-carboxylate (**16a**)

3-Fluoro-4-methoxyaniline (60 mg, 0.43 mmol), palladium(II) acetate (14 mg, 0.06 mmol), cesium carbonate (88 mg, 0.64 mmol) and 2-(dicyclohexylphosphino)-2′,4′,6′-tri-i-propyl-1,1′-biphenyl (69 mg, 0.13 mmol) were added to a solution of compound **15** (100 mg, 0.21 mmol) in 1,4-dioxane (30 mL) under the protection of nitrogen. The mixture was stirred at 100 °C for 2 h and monitored by TLC. Upon completion, the solution was concentrated under reduced pressure to provide the crude product, which was purified with silica gel chromatography (1:5 *v*/*v* ethyl acetate/petroleum ether) to afford a white solid (104 mg, 80%). mp 142.3–145.6 °C; ^1^H NMR (400 MHz, DMSO-*d_6_*) *δ* 8.13 (s, 1H), 7.83 (d, *J* = 8.8 Hz, 1H), 7.66 (d, *J* = 8.0 Hz, 2H), 7.22 (d, *J* = 8.0 Hz, 2H), 6.99 (t, *J* = 9.2 Hz, 1H), 6.90 (s, 1H), 6.88–6.83 (m, 1H), 6.80–6.76 (m, 1H), 5.74 (s, 2H), 5.12–5.06 (m, 1H), 4.97 (s, 2H), 4.81 (t, *J* = 5.6 Hz, 1H), 3.77 (s, 3H), 1.24 (d, *J* = 6.0 Hz, 6H). ^13^C NMR (100 MHz, DMSO-*d_6_*) *δ* 161.75, 153.74, 151.33, 144.23, 142.91, 141.46, 141.35, 140.00, 138.03, 137.95, 128.37, 128.05, 127.37, 126.32, 125.90, 125.86, 123.70, 123.13, 120.90, 115.88, 113.95, 113.82, 113.79, 106.46, 106.25, 95.76, 68.44, 57.12, 56.05, 47.75, 22.10. ESI-HRMS (*m*/*z*): calcd C_28_H_26_F_4_N_2_O_4_ for [M − H]^−^ 529.1829; found 529.1761. R_T,HPLC_ = 13.01 min. The same procedure was also followed for the synthesis of **16b**, **19a** and **19b**.

#### 4.5.6. Isopropyl 6-((2,4-difluorophenyl)amino)-3-(((4-(trifluoromethyl)benzyl)oxy) methyl)-1*H*-indole-2-carboxylate (**16b**)

White solid, 85% yield. mp 149.7–152.5 °C; ^1^H NMR (400 MHz, DMSO-*d_6_*) *δ* 8.04 (s, 1H), 7.83 (d, *J* = 8.4 Hz, 1H), 7.69 (d, *J* = 8.4 Hz, 2H), 7.29–7.18 (m, 4H), 6.94(t, *J* = 8.4 Hz, 1H) 6.85(d, *J* = 8.8 Hz, 1H), 6.80 (s, 1H), 5.72 (s, 2H), 5.11–5.05 (m, 1H), 4.97 (t, *J* = 3.6 Hz, 2H), 1.23 (d, *J* = 6.4 Hz, 6H). ^13^C NMR (100 MHz, DMSO-*d_6_*) *δ* 161.73, 144.19, 143.25, 139.99, 128.36, 128.04, 127.38, 126.32, 125.87, 123.49, 123.18, 122.34, 120.99, 113.50, 111.83, 111.61, 105.27, 105.04, 104.77, 96.01, 68.44, 56.01, 47.77, 22.07. ESI-HRMS (*m*/*z*): calcd C_27_H_23_F_5_N_2_O_3_ for [M + Na]^+^ 541.1527; found 541.1504. R_T,HPLC_ = 16.74 min.

#### 4.5.7. Synthesis of 6-((3-fluoro-4-methoxyphenyl)amino)-3-(((4-(trifluoromethyl) benzyl)oxy)methyl) -1*H*-indole-2-carboxylic acid (**17a**)

Sodium hydroxide (28 mg, 0.71 mmol) was added to a solution of **15** (100 mg, 0.24 mmol) in mixed solution of methanol and water (4 mL, 3:1 methanol/water), and the reaction was stirred at 80 °C for 1.5 h. The mixture was then cooled to room temperature, brought to a pH of 6 by adding acetic acid, extracted with ethyl acetate (15 mL × 3) and concentrated under reduced pressure to provide the crude product. The residue was purified with silica gel chromatography (1:2 *v*/*v* ethyl acetate/petroleum ether) to afford a yellow solid (44 mg, 45%). mp 134.8–136.3 °C; ^1^H NMR (400 MHz, DMSO-*d_6_*) *δ* 8.12 (s, 1H), 7.69–7.65 (m, 3H), 7.21 (d, *J* = 8.0 Hz, 2H), 7.00–6.95 (m, 1H), 6.89–6.73 (m, 4H), 5.81 (s, 2H), 4.92 (S, 2H), 3.77 (s, 3H). ^13^C NMR (100 MHz, DMSO-*d_6_*) *δ* 163.67, 144.25, 142.74, 141.41, 141.31, 139.75, 138.06, 137.96, 127.41, 125.95, 125.91, 123.04, 121.42, 121.10, 114.24, 113.73, 113.70, 106.37, 106.16, 96.01, 66.03, 57.81, 57.13. ESI-HRMS (*m*/*z*): calcd C_25_H_20_F_4_N_2_O_4_ for [M − H]^−^ 487.1359; found 487.1303. R_T,HPLC_ = 7.34 min. The same procedure was also followed for the synthesis of **17b**, **20a**, **20b** and **21**.

#### 4.5.8. 6-((2,4-difluorophenyl)amino)-3-(((4-(trifluoromethyl)benzyl)oxy)methyl)-1*H*-indole-2-carboxylic acid (**17b**)

Yellow solid, 50% yield. mp 157.3–160.5 °C; ^1^H NMR (400 MHz, DMSO-*d_6_*) *δ* 7.89 (s, 1H), 7.66 (d, *J* = 8.0 Hz, 3H), 7.22 (d, *J* = 8.0 Hz, 3H), 7.17–7.11(m, 1H), 6.9–6.78 (m, 3H), 5.88 (s, 2H), 4.87 (s, 2H). ^13^C NMR (100 MHz, DMSO-*d_6_*) *δ* 172.63, 165.09, 157.68, 155.41, 155.30, 152.72, 144.83, 141.52, 138.66, 128.77, 128.62, 128.05, 127.73, 127.61, 126.18, 125.82, 125.78, 123.48, 122.15, 120.96, 113.24, 111.71, 111.49, 105.24, 105.00, 104.73, 97.10, 55.68, 47.13. ESI-HRMS (*m*/*z*): calcd C_24_H_17_F_5_N_2_O_3_ for [M − H]^−^ 475.1159; found 475.1175. R_T,HPLC_ = 13.94 min.

#### 4.5.9. Isopropyl 6-bromo-3-(((2-fluorobenzyl)oxy)methyl)-1*H*-indole-2-carboxylate (**18**)

White solid, 89% yield. mp 154.6–158.9 °C; ^1^H NMR (400 MHz, DMSO-*d_6_*) *δ* 7.93 (d, *J* = 8.4 Hz, 1H), 7.86 (s, 1H), 7.28 (d, *J* = 8.4 Hz, 2H), 7.22–7.18 (m, 1H), 7.03 (t, *J* = 7.2 Hz, 1H), 6.49 (t, *J* = 7.6 Hz, 1H), 5.80 (s, 2H), 5.12–5.05 (m, 1H), 4.98 (s, 2H), 1.20 (d, *J* = 6.4 Hz, 6H). ^13^C NMR (100 MHz, DMSO-*d_6_*) *δ* 161.36, 143.86, 139.30, 127.18, 125.93, 125.89, 125.62, 125.60, 125.40, 124.56, 124.06, 123.38, 119.13, 114.00, 69.19, 55.71, 47.90, 21.92. ESI-HRMS (*m*/*z*): calcd C_20_H_19_BrFNO_3_ for [M + Na]^+^ 442.0430, 444.0410; found 442.0438, 444.0415. R_T,HPLC_ = 15.44 min.

#### 4.5.10. Isopropyl 6-((3-fluoro-4-methoxyphenyl)amino)-3-(((2-fluorobenzyl)oxy) methyl)-1*H*-indole-2-carboxylate (**19a**)

White solid, 79% yield. mp 141.2–144.5 °C; ^1^H NMR (400 MHz, DMSO-*d_6_*) *δ* 8.14 (s, 1H), 7.81 (d, *J* = 8.8 Hz, 1H), 7.31–7.25 (m, 1H), 7.19 (t, *J* = 8.8 Hz, 1H) 7.07–7.00 (m,2H), 6.94 (s, 1H), 6.89–6.79 (m, 3H), 6.64 (t, *J* = 7.6 Hz, 1H), 5.70 (s, 2H), 5.13–5.06 (m,1H), 4.97 (d, *J* = 5.6 Hz, 2H), 4.80 (t, *J* = 5.2 Hz, 1H), 3.78 (s, 3H), 1.24 (d, *J* = 6.4 Hz, 6H). ^13^C NMR (100 MHz, DMSO-*d_6_*) *δ* 161.71, 161.19, 158.76, 153.77, 151.35, 142.93, 141.48, 141.37, 140.07, 137.92, 129.51, 129.43, 128.39, 128.34, 126.32, 126.11, 125.97, 125.01, 123.62, 123.24, 120.81, 115.81, 115.60, 113.88, 106.58, 106.38, 95.46, 68.36, 57.17, 55.99, 22.06. ESI-HRMS (*m*/*z*): calcd C_27_H_26_F_2_N_2_O_4_ for [M + Na]^+^: 503.1758; found: 503.1766. R_T,HPLC_ = 6.33 min, purity > 99%.

#### 4.5.11. Isopropyl 6-((2,4-difluorophenyl)amino)-3-(((2-fluorobenzyl)oxy)methyl)-1*H*-indole-2-carboxylate (**19b**)

White solid, 68% yield. mp 171.8–175.3 °C; ^1^H NMR (400 MHz, DMSO-*d_6_*) *δ* 7.89 (s, 1H), 7.80 (d, *J* = 8.8 Hz, 1H), 7.28–7.17 (m, 4H), 7.04 (t, *J* = 7.2 Hz, 1H), 6.95 (t, *J* = 8.8 Hz, 1H), 6.85–6.80 (m, 2H), 6.61 (t, *J* = 7.6 Hz, 1H), 5.66 (s, 2H), 5.12–5.06 (m, 1H), 4.97 (d, *J* = 5.6 Hz, 2H), 4.80 (t, *J* = 5.6 Hz, 1H), 1.23 (d, *J* = 6.4 Hz, 6H). ^13^C NMR (100 MHz, DMSO-*d_6_*) *δ* 161.70, 161.18, 158.75, 143.26, 140.02, 129.49, 129.41, 128.36, 128.32, 126.30, 126.09, 125.94, 125.00, 123.42, 123.29, 120.88, 115.78, 115.57, 113.44, 111.85, 111.64, 105.30, 105.04, 104.80, 95.70, 68.38, 55.97, 22.04. ESI-HRMS (*m*/*z*): calcd C_26_H_23_F_3_N_2_O_3_ for [M + Na]^+^: 491.1558; found 491.1538. R_T,HPLC_ = 7.32 min.

#### 4.5.12. 6-((3-fluoro-4-methoxyphenyl)amino)-3-(((2-fluorobenzyl)oxy)methyl)-1*H*-indole-2-carboxylic acid (**20a**)

Yellow solid, 47% yield. mp 169.9–173.7 °C; ^1^H NMR (400 MHz, DMSO-*d_6_*) *δ* 8.13 (s, 1H), 7.78 (d, *J* = 8.4 Hz, 1H), 7.30–7.17 (m, 2H), 7.07–7.00 (m, 2H), 6.91 (s, 1H), 6.86–6.78 (m, 3H), 6.64 (s, 1H), 5.76 (s, 2H), 4.97 (s, 2H), 3.78 (s, 3H). ^13^C NMR (100 MHz, DMSO-*d_6_*) *δ* 163.76, 161.22, 158.79, 153.77, 151.35, 142.71, 141.41, 141.30, 139.91, 138.10, 138.02, 128.51, 128.47, 126.17, 126.03, 125.84, 125.00, 123.76, 123.42, 120.85, 115.94, 113.75, 106.44, 106.24, 95.60, 57.19, 55.89. ESI-HRMS (*m*/*z*): calcd C_24_H_20_F_2_N_2_O_4_ for [M − H]^−^: 437.1391; found 437.1301. R_T,HPLC_ = 9.81 min.

#### 4.5.13. 6-((2,4-difluorophenyl)amino)-3-(((2-fluorobenzyl)oxy)methyl)-1*H*-indole-2-carboxylic acid (**20b**)

Yellow solid, 52% yield. mp 178.5–181.8 °C; ^1^H NMR (400 MHz, CD_3_OD) *δ* 7.71 (d, *J* = 8.8 Hz, 1H), 7.23–7.13 (m, 2H), 7.10–7.04 (m, 1H), 6.98–6.94 (m, 2H), 6.93 (s, 1H), 6.88 (dd, *J* = 8.8, 2.0 Hz, 1H), 6.81–6.77 (m, 2H), 6.63 (t, *J* = 9.2 Hz, 1H), 5.79 (s, 2H), 5.09 (s, 2H). ^13^C NMR (100 MHz, CD_3_OD) *δ* 162.60, 160.17, 144.21, 141.06, 129.88, 129.80, 129.24, 129.20, 127.12, 126.98, 125.53, 125.39, 125.35, 122.90, 122.10, 116.13, 115.92, 114.82, 111.96, 111.74, 105.40, 105.13, 104.89, 96.85, 56.44, 42.42. ESI-HRMS (*m*/*z*): calcd C_23_H_17_F_3_N_2_O_3_ for [M − H]^−^: 425.1191; found 425.1117. R_T,HPLC_ = 13.12 min.

#### 4.5.14. 6-bromo-3-(((2-fluorobenzyl)oxy)methyl)-1*H*-indole-2-carboxylic acid (**21**)

White solid, 48% yield. mp 174.3–178.0 °C, ^1^H NMR (400 MHz, DMSO-*d_6_*) *δ* 7.91 (d, *J* = 8.8 Hz 1H), 7.86 (d, *J* = 1.6 Hz 1H), 7.28 (d, *J* = 8.8 Hz 2H), 7.24–7.19 (m, 1H), 7.01 (t, *J* = 7.6 Hz, 1H), 6.44 (t, *J* = 7.6 Hz, 1H), 5.86 (s, 2H), 4.98 (s, 2H). ^13^C NMR (100 MHz, DMSO-*d_6_*) *δ* 163.24, 161.28, 158.84, 139.26, 129.54, 127.93, 127.69, 125.95, 125.81, 125.77, 125.07, 124.24, 123.86, 120.61, 118.83, 115.91, 115.70, 114.19, 65.63, 57.85. ESI-HRMS (*m*/*z*): calcd C_17_H_13_BrFNO_3_ for [M − H]^−^: 375.9985, 377.9964; found 375.9975, 377.9953. R_T,HPLC_ = 6.84 min. 

### 4.6. Biochemistry

#### 4.6.1. Cell Lines, and Culture Conditions

All the reagents and chemicals were purchased from commercial sources. HIV-1 Integrase Assay Kit was purchased from Abnova. MT-4 cells were cultured in RPMI-1640 media and equipped with 10% FBS (Gibco, Sidney, Australia), 100 units/mL penicillin and 100 units/mL streptomycin at 37 °C in a humidified atmosphere (5% CO_2_ and 95% air). 

#### 4.6.2. Strand Transfer Inhibition Assay

The strand transfer assay was performed using an HIV-1 integrase assay kit (Abnova, Taipei, CN) according to the manufacturer’s instructions. Each sample was dissolved in DMSO and diluted to a final desired concentration (80, 40, 20, 10, 5 μM) with reaction buffer. Briefly, 100 μL of DS DNA solution was added to each well and incubated for 30 min at 37 °C, and the liquid was removed from the wells and washed 3 times with wash buffer. Then, 200 μL of blocking buffer was added to each well and incubated for 30 min in a 37 °C incubator. Followed by the aspiration of the liquid, each well was washed 3 times with reaction buffer, and 100 μL integrase enzyme solution was added to each well and incubated under similar conditions. Sequentially, 50 μL reaction buffer containing different concentrations of samples was added to each well and incubated for 5 min at room temperature. To each well was added 50 μL of TS DNA solution, and the reactions were mixed gently and incubated for 30 min at 37 °C. Then, each well was washed 3 times with wash solution, incubated with 100 μL HRP antibody and incubated for 30 min at 37 °C. After washing the plate under the same conditions, 100 μL TMB peroxidase substrate solution was added to each well and incubated at room temperature for 10 min. To terminate the reaction, 100 μL TMB stop solution was directly added to each well and the absorbance of the sample was determined using a plate reader (ELx800UV, Bio-Tek Instrument Inc., Winooski, VT, USA) at 450 nm.

#### 4.6.3. Cytotoxicity Assay

The cytotoxicity of the title compounds was tested through MTT analysis on MT-4 cells. Cells were seeded in 96-well plates with a density of 50,000 cells/well and coincubated with serial diluted compounds (2.5, 5, 10, 20, 40, and 80 μM) at 37 °C for five days. MTT (Sigma) was then added to each well to a final concentration of 0.5 mg/mL and incubated for 2 h. Subsequently, the medium was decanted, and 150 μL of DMSO was added to each well. The absorbance was obtained using a microplate reader at 570 nm, and all measurements were conducted three times under the same condition. Finally, 50% cytotoxicity concentration (CC50) values were calculated by Graphpad Prism 8.

## Data Availability

Data are contained within the article and Appendix A.

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
