# Peer review of "The Discovery of Indole-2-carboxylic Acid Derivatives as Novel HIV-1 Integrase Strand Transfer Inhibitors"

_molecules, 2023, doi:10.3390/molecules28248020_

Round 1
Reviewer 1 Report
Comments and Suggestions for Authors
The manuscript “Discovery of indole-2-carboxylic acid derivatives as novel HIV-1 integrase strand transfer inhibitors” is devoted to the rational design of series (11 compounds) of indole-2-carboxylic acid derivatives as HIV-1 integrase strand transfer inhibitors. The research is comprehensive and is of high practical significance. The authors have found one compound to be a promising HIV-1 INSTI which is very interesting for the development of new anti-HIV drugs.
In my opinion, this manuscript suits to the scope of Molecules. I recommend that after minor revision, it can be accepted.
Some comments for the authors:
1. Line 172: CS2CO3?
2. For each reported earlier compound, a reference should be added and the reported earlier m.p.
Author Response
Comment 1. Some comments for the authors: 1. Line 172: CS2CO3?
Response: Revised.
Page 6, line 174:
“…(e) Pd(OAc)2, Cs2CO3, Xphos…”
Comment 2. For each reported earlier compound, a reference should be added and the reported earlier m.p.
Response: Thanks for the helpful suggestion. Regretfully, no reference could be found for these compounds in various chemical databases (e.g., SCIFinder, ChemSpider, Reaxys), except chemical suppliers. For similar virtual screening researches, no reference was found for the obtained compounds either (J. Chem. Inf. Model., 2022, 62, 1, 116–128; BMC Bioinformatics, 2011, 12(Suppl 13): S24; Molecules, 2020, 25, 3193). However, the m.p. of each compound has been tested and added in the Table 1 as follows.
Page 4, line 110:
Table 1. Binding energies and HIV-1 integrase inhibitory effect of 10 compounds from virtual screening.
|
Compd. |
HIT ID |
m.p. (°C)a |
Lowest binding energy (kcal/mol) |
Highest binding energy (kcal/mol) |
Most binding energy (kcal/mol) |
IC50 (μM)b |
|
1 |
HIT101057921 |
173.4−175.3 |
–15.8 |
–11.4 |
–15.8 |
39.06 ± 1.21 |
|
2 |
HIT104550697 |
113.0−116.7 |
–15.3 |
–11.3 |
–15.3 |
33.01 ± 1.29 |
|
3 |
HIT104315479 |
159.7−163.2 |
–15.1 |
–14.5 |
–15.1 |
12.41 ± 0.07 |
|
4 |
HIT100811644 |
271.4−279.0 |
–13.4 |
–11.0 |
–13.4 |
18.52 ± 1.06 |
|
5 |
HIT106924020 |
164.7−167.8 |
–13.2 |
–8.8 |
–13.2 |
47.44 ± 1.45 |
|
6 |
HIT105485118 |
235.3−241.2 |
–13.2 |
–11.7 |
–13.1 |
22.63 ± 0.58 |
|
7 |
HIT105499167 |
147.0−148.2 |
–13.2 |
–9.4 |
–11.9 |
21.45 ± 0.26 |
|
8 |
HIT106066563 |
171.5−176.8 |
–13.1 |
–8.5 |
–12.7 |
28.16 ± 1.07 |
|
9 |
HIT101131099 |
239.0−240.1 |
–13.1 |
–9.7 |
–11.7 |
32.75 ± 0.83 |
|
10 |
HIT102924146 |
139.9−143.6 |
–12.9 |
–11.4 |
–12.9 |
22.25 ± 0.43 |
|
RAL |
-c |
154.3-157.9 |
–13.6 |
–8.9 |
–12.0 |
0.08 ± 0.04 |
αMelting point.
bConcentration required to inhibit the in vitro strand-transfer step of integrase by 50%. IC50 values are average for three independent experiments.
cNot applicable.

Reviewer 2 Report
Comments and Suggestions for Authors
In this manuscript, the authors discovered a couple of indole-2-carboxylic derivatives as novel integrase strand transfer inhibitors. The authors established this finding through a combination of a set of molecular docking-based virtual screening and syntheses of several derivatives by using a unique backbone structure with similar pharmacophore to the approved integrase strand transfer inhibitors such as raltegravir (RAL).
Although this study was rationally designed and could result in useful information in terms of development of novel integrase strand transfer inhibitors, the reviewer think that the authors should demonstrate antiviral activity data using some HIV strains and MT-4 cells, in which the cytotoxicity of the indole-2-carboxylic derivatives was shown.
(minor point)
The data for RAL should be included in Table 1.
Author Response
Comment 1. Although this study was rationally designed and could result in useful information in terms of development of novel integrase strand transfer inhibitors, the reviewer think that the authors should demonstrate antiviral activity data using some HIV strains and MT-4 cells, in which the cytotoxicity of the indole-2-carboxylic derivatives was shown.
Response: We really appreciate your helpful advice, and we are also very interested in the antiviral effect of these compounds. However, due to the high requirements for conducting anti-HIV-assays, biological safety protection third-level laboratory (P3) is needed to carried out the experiment. Especially after the COVID-19 epidemic, the study involving antiviral evaluations have been managed more strictly, making it impossible for our labs to conduct the antiviral evaluation. Therefore, we contacted several labs of Center for Disease Control (CDC) which could perform HIV-1 investigations (such as Beijing Center for Disease Prevention and Control, and Guizhou Center for Disease Control and Prevention), whereas all these departments only test samples obtained from patients with AIDS instead of small molecule compounds.
There are very few research institutes in our country that can conduct antiviral assays of small molecule compound, and we have contacted several of them (e.g., Kunming Institute of Zoology and Wuhan Institute of Virology). Regretfully, the cost is too expensive for our lab (>50,000 RMB per sample, or >7,000 US dollars per sample). All these factors make it very hard for us to carry out the antiviral activity test.
Comment 2. The data for RAL should be included in Table 1.
Response: According to your suggestion, the data for RAL has been included in Table 1.
Page 4, line 110:
Table 1. Binding energies and HIV-1 integrase inhibitory effect of 10 compounds from virtual screening.
|
Compd. |
HIT ID |
m.p. (°C)a |
Lowest binding energy (kcal/mol) |
Highest binding energy (kcal/mol) |
Most binding energy (kcal/mol) |
IC50 (μM)b |
|
1 |
HIT101057921 |
173.4−175.3 |
–15.8 |
–11.4 |
–15.8 |
39.06 ± 1.21 |
|
2 |
HIT104550697 |
113.0−116.7 |
–15.3 |
–11.3 |
–15.3 |
33.01 ± 1.29 |
|
3 |
HIT104315479 |
159.7−163.2 |
–15.1 |
–14.5 |
–15.1 |
12.41 ± 0.07 |
|
4 |
HIT100811644 |
271.4−279.0 |
–13.4 |
–11.0 |
–13.4 |
18.52 ± 1.06 |
|
5 |
HIT106924020 |
164.7−167.8 |
–13.2 |
–8.8 |
–13.2 |
47.44 ± 1.45 |
|
6 |
HIT105485118 |
235.3−241.2 |
–13.2 |
–11.7 |
–13.1 |
22.63 ± 0.58 |
|
7 |
HIT105499167 |
147.0−148.2 |
–13.2 |
–9.4 |
–11.9 |
21.45 ± 0.26 |
|
8 |
HIT106066563 |
171.5−176.8 |
–13.1 |
–8.5 |
–12.7 |
28.16 ± 1.07 |
|
9 |
HIT101131099 |
239.0−240.1 |
–13.1 |
–9.7 |
–11.7 |
32.75 ± 0.83 |
|
10 |
HIT102924146 |
139.9−143.6 |
–12.9 |
–11.4 |
–12.9 |
22.25 ± 0.43 |
|
RAL |
-c |
154.3-157.9 |
–13.6 |
–8.9 |
–12.0 |
0.08 ± 0.04 |
αMelting point.
bConcentration required to inhibit the in vitro strand-transfer step of integrase by 50%. IC50 values are average for three independent experiments.
cNot applicable.

Round 2
Reviewer 2 Report
Comments and Suggestions for Authors
The reviewer confirmed the author's revision and response.